# Deep Reinforcement Learning for Flipper Control of Tracked Robots in Urban Rescuing Environments

Hainan Pan , Xieyuanli Chen , Junkai Ren *, Bailiang Chen, Kaihong Huang, Hui Zhang and Huimin Lu

College of Intelligence Science and Technology, National University of Defense Technology, Changsha 410073, China; phn@nudt.edu.cn (H.P.); xieyuanli.chen@nudt.edu.cn (X.C.); chenbailiang12@nudt.edu.cn (B.C.); kaihong.huang@nudt.edu.cn (K.H.); huizhang_nudt@nudt.edu.cn (H.Z.); lhmnew@nudt.edu.cn (H.L.)
* Correspondence: jk.ren@nudt.edu.cn

**Abstract:** Tracked robots equipped with flippers and LiDAR sensors have been widely used in urban search and rescue. Achieving autonomous flipper control is important in enhancing the intelligent operation of tracked robots within complex urban rescuing environments. While existing methods mainly rely on the heavy work of manual modeling, this paper proposes a novel Deep Reinforcement Learning (DRL) approach named ICM-D3QN for autonomous flipper control in complex urban rescuing terrains. Specifically, ICM-D3QN comprises three modules: a feature extraction and fusion module for extracting and integrating robot and environment state features, a curiosity module for enhancing the efficiency of flipper action exploration, and a deep Q-Learning control module for learning robot-control policy. In addition, a specific reward function is designed, considering both safety and passing smoothness. Furthermore, simulation environments are constructed using the Pymunk and Gazebo physics engine for training and testing. The learned policy is then directly transferred to our self-designed tracked robot in a real-world environment for quantitative analysis. The consistently high performance of the proposed approach validates its superiority over hand-crafted control models and state-of-the-art DRL strategies for crossing complex terrains.

**Keywords:** Deep Reinforcement Learning; rescue robot; intelligent systems; robot control

## 1. Introduction

In the process of solving the Urban Search and Rescue (USAR) problem of utilizing ground mobile robots, growing attention has been paid to trajectory planning and navigation in complex environments, such as multilayered or rugged terrains [1]. However, one of the main challenges still lies in designing effective strategies to control robots crossing obstacles safely and effectively. Tracked robots equipped with four flippers exhibit exceptional terrain traversal capabilities, facilitating efficient navigation through uneven terrain during USAR missions [2,3]. While adding multiple flippers enhances the traversability of the tracked robots, it also introduces a high degree of control freedom. In complex terrain environments, relying solely on manual control can impose a significant cognitive burden and increase the time required for terrain traversal tasks, potentially impacting the rescue success rate [4,5]. Consequently, achieving autonomous terrain traversal is paramount in augmenting the intelligent operation of tracked robots in USAR tasks.

Since articulated tracked robots can adjust their morphology for safe and efficient obstacle crossing through flipper movements, extensive research has focused on addressing the issue of autonomous flipper control. Early studies [6–8] primarily centered on analyzing the robot kinematics with specific terrain structures, incorporating additional constraints and simplifications that limit the extensibility of the traversal model. In contrast, recent research based on Deep Reinforcement Learning (DRL) [9] has allowed researchers to adopt a more practical approach to devise strategies for traversing obstacles, leveraging the fitting capabilities of Deep Learning (DL) methods and the optimization capabilities

of Reinforcement Learning (RL) to solve problems more efficiently. However, the optimal flipper actions that satisfy the constraints are sparse in large action spaces. Inefficient action exploration efficiency reduces DRL algorithms' training efficiency and control, and the recently proposed Intrinsic Curiosity Module (ICM) [10] helps enhance the algorithmic action exploration efficiency.

In this article, we aim to investigate DRL for the autonomous control of flippers during obstacle traversal by tracked robots to enhance efficiency and reduce the operational burden on human operators, as shown in Figure 1. The key contribution of this work lies in developing a DRL-based autonomous flipper control algorithm. To be more specific, we design a novel DRL network architecture named ICM-D3QN, consisting of three main modules. The first module fuses the states of the robot and the environment and extracts compact features. Meanwhile, the second module enhances the algorithm's action exploration efficiency by directing unexplored flipper actions through an ICM. We finally apply the Double Dueling Deep Q-Learing Network (D3QN) [11] to provide the proper flipper controls, prioritizing the satisfaction of prior knowledge and smoothness constraints. To enhance the algorithm's generalization capability, we adopt Domain Randomization (DR) [12–15] techniques, and the traversal strategy is learned in multiple simulated terrain environments with varying types, sizes, and noise levels. After integrating our algorithm, our flipper-based tracked robots acquire the capability to navigate various types of terrain and overcome obstacles effectively. The experimental results show that the proposed approach outperforms manual modeling [16] and Mitriakov's up-and-down staircase strategy [17], exhibiting improved traversal capacity across different environments.

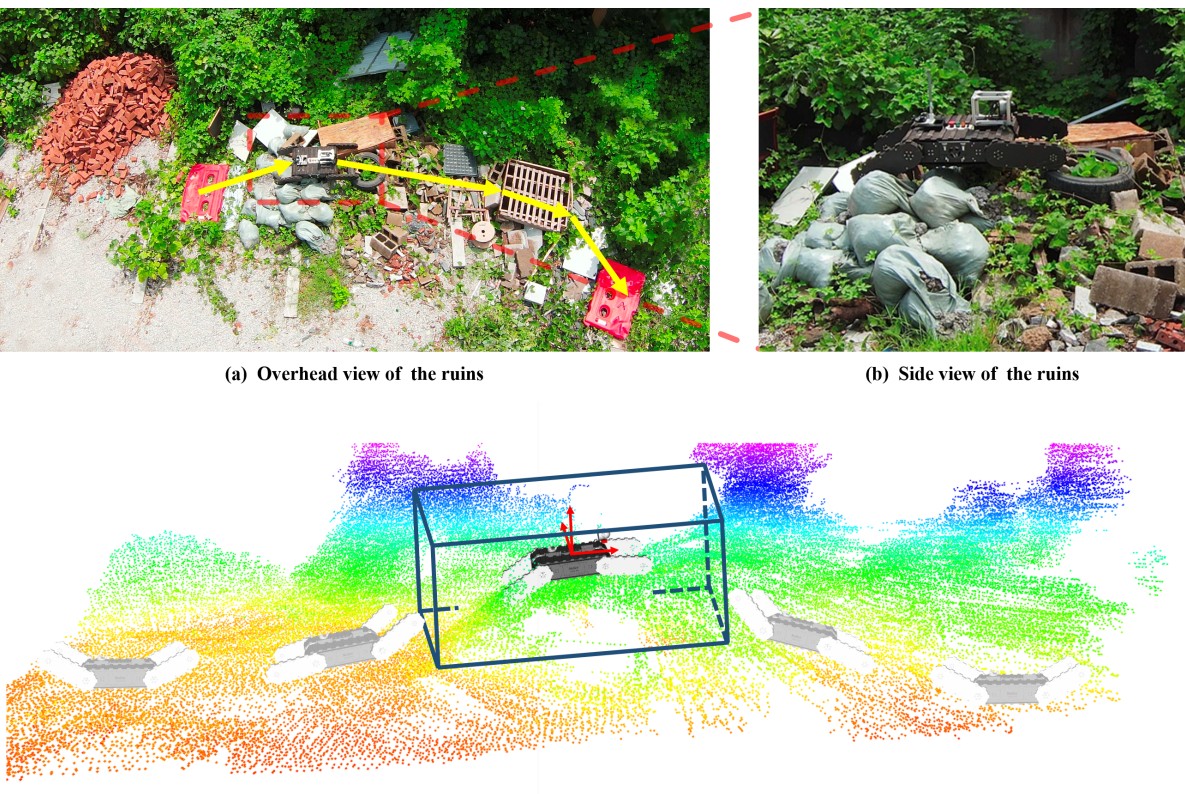

(a) Overhead view of the ruins

(b) Side view of the ruins

(c) ICM-D3QN flipper autonomous control algorithm with local terrain information and ontology state as inputs

**Figure 1.** Our NuBot-Rescue tracked robot runs in urban search and rescue environments. It has four flippers and LiDAR and IMU sensors, showcasing remarkable capabilities in traversing various terrains: The yellow path in (**a**) is the direction in which the robot is traveling; the red dotted line refers to the locally enlarged view of the robot crossing obstacles, as shown in (**b**); and the blue cube in (**c**) is the range of local terrain information.

To sum up, the main contributions of this article are threefold:

- A novel D3QN-based flipper autonomy control algorithm is proposed, which autonomously regulates the movement of flippers for obstacle-crossing tasks using local terrain information and ontology state.
- An ICM module has been introduced to enhance the efficiency of action exploration during the D3QN training process. Ablation experiments demonstrate the algorithm's convergence capability and obstacle-crossing performance have been significantly optimized.
- The DR technique is employed to train the flipper control strategy in the Pymunk simulation encompassing a diverse range of terrain variations and directly transfer it to the more realistic Gazebo simulation and real-world USAR ruins for validation. The results show that our proposed algorithm has a better generalization ability, achieving better obstacle-crossing smoothness than the state-of-the-art methods.

## 2. Related Work

The autonomous control of articulated tracked robots has been studied for decades. Early works [6–8,18] focused on developing terrain traversal models for the robot in specific terrains, including geometric, kinematic, and even dynamic models. Li et al. [18] analyzed the geometric correlation between centroid displacement and the traversal of steps and stairs by a double-flipper tracked robot. There are also works [6–8] developing dynamic models for the sequential locomotion of robots ascending and descending steps and stairs. Such works devise motion strategies for the front and rear flippers based on aligning the robot chassis with the terrain envelope during manual operation as closely as possible. To ensure stability, inappropriate flipper configurations were identified by evaluating the robot's normalized energy stability margin. These models were analyzed to determine the proper poses of the robot and used for designing the control strategies accordingly. However, the analysis of these models was limited to simple and single terrains. When constructing and tackling complex environments, existing works usually fail.

Machine learning technology has rapidly developed in recent years, and researchers have turned their attention to robot control methods based on learning-based techniques. Paolo et al. [19] initially employed a comprehensive end-to-end DRL approach to address the challenge of autonomous flipper control. They utilized a Convolutional Neural Network (CNN) to extract depth image features from the robot's front and rear perspectives. These features and the robot's state information were incorporated into the Deep Deterministic Policy Gradient (DDPG) algorithm framework for training purposes. However, the high cost associated with image-based training could have helped to achieve satisfactory outcomes. Works by Mitriakov et al. [20,21] optimized the mechanical arm's and chassis's overall stability by incorporating it into the reward function. Their methods solely focused on safety constraints, disregarding the potential for enhanced and expedited obstacle traversal. However, their recent research marked the pioneering implementation of DR techniques in training articulated tracked robots [17]. Their investigation primarily revolved around 3D navigation within structured indoor environments, where DR was embodied in mazes and staircases set with multiple parameters. Contrary to their research focus, our proposal posits that applying DR techniques can bolster the robot's autonomous prowess in overcoming obstacles and augment its adaptability in intricate surroundings. Instead, Zimmermann et al. [22] used real terrain traversal data as the expectation, and the feature with the smallest residual in Robot Terrain Interaction (RTI) was extracted by DL. The Q-learning method was then used to learn the strategy of switching among five predefined flippers from the feature. Azayev et al. [23] used data from manual teleoperation, and a state machine network based on Imitation Learning (IL) was proposed to optimize the unreasonable flipper action switching in [22]. Their approach considers the beneficial impact of manual operational expertise on algorithmic control. However, it necessitates substantial data acquisition and learning costs. Furthermore, the algorithm's control effectiveness switches between predefined states and actions, thereby failing to exploit flippers' constructive capabilities during obstacle traversal.

From the above research, DRL can better fit the nonlinear contact model between the robot and the terrain by combining DL with RL. This allows for obtaining the action strategy with multiple index constraints, which has great advantages in solving the high-dimensional and complex problem of tracked robots crossing obstacles. Nevertheless, attributable to the exorbitant expenses associated with data acquisition and training, achieving substantial advancements in current research poses a major challenge. Moreover, numerous studies focusing solely on security constraints lose sight of the beneficial impact of artificial experiential knowledge on algorithms, resulting in a sluggish and incongruous obstacle-traversal effect. In addition, DRL relies heavily on the quantity and quality of experience samples, which often leads to a decrease in the quality of samples due to insufficient action exploration, thus affecting the training result. In the field of DRL, the generalized mechanism of the curiosity module is to improve a robot's learning efficiency and performance by encouraging it to try out actions and explore new states during the learning process.

Current research on the curiosity module includes information gain, goal-directed, and unsupervised exploration. The first module was usually realized by a Bayesian approach [24], which selected the actions with the largest confidence, i.e., uncertainty, for exploration by estimating the confidence of the state transferred after all the executed actions of the robot. It was a supervised learning method, requiring labeled experience and data for the quantitative evaluation of features. The second module generally sets the goal of exploration [25,26], such as the number of times a certain state is explored, to encourage the robot to explore toward the goal state with fewer explorations. This approach relied on setting goals, which depended on good prior knowledge, resulting in a more limited strategy. On the other hand, curiosity modules for unsupervised exploration encoded the state space through a network such as a self-encoder, thereby encouraging the robot to explore state features that were different from its current experience. This approach belonged to the internally driven curiosity module [10], which could discover unexplored information by driving the robot to explore new states through the error of self-encoded state features without external incentive signals. Combining it with the reward incentive of DRL methods could assist the robot in learning a better action strategy in the presence of sufficient exploration.

Unlike existing DRL methods for flipper control, our proposal applies to real-time smooth obstacle crossing in various urban complex environments. We construct various terrain scenarios in the 2D Pymunk simulation with lower sample acquisition costs. Additionally, we devise a reward function that considers prior knowledge and smoothness metrics to guide the training process. By integrating ICM, we significantly enhance the exploration efficiency of the extensive state-action space during training. Consequently, we propose an ICM-D3QN flipper autonomy control algorithm tailored for discrete motion space, enabling the robot to achieve improved convergence and traversal capabilities across diverse terrains. We successfully implement this algorithm in both a 3D simulation environment and a real-world setting, subjecting it to rigorous testing on challenging and complex urban scenes. The results demonstrate its ability to facilitate the articulated tracked robot's smooth and swift traversal of obstacles in real time.

## 3. Methodology

### 3.1. Problem Formulation

We employ our self-designed NuBot-Rescue robot as the experimental platform, as shown in Figure 1. This platform offers advantageous central symmetry properties, and its components, namely the tracks and flippers, can be controlled independently. In this study, we assume that human operators or path-planning algorithms are responsible for controlling the rotation of the robot tracks, while the developed autonomous flipper algorithm governs the motion of the flippers.

In real-world scenarios involving traversing complex terrains, minimizing robot instability, such as side-slipping, is important. To achieve this, human operators typically

align the robot's forward direction with the undulating terrain of obstacles [27], employing similar measures for both the left and right flippers. Building upon this premise, we project the terrain outline and robot shape onto the robot's lateral side. An example of the robot's and terrain interaction is shown in Figure 2. Our approach suits environments with minor left-to-right fluctuations and significant up-and-down fluctuations.

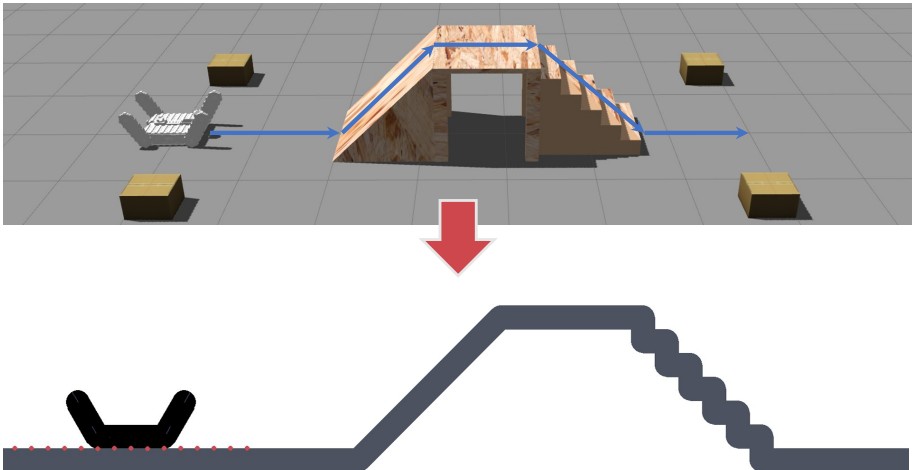

**Figure 2.** Terrain and robot model simplification in this paper. The blue arrow is the hypothetical path human remote control chooses to reduce instability. The red dots represent the terrain points in the local perception domain.

In this article, we use DRL to develop an autonomous control system for the flippers of a tracked robot. Specifically, we formulate the problem as a Markov Decision Processes (MDP) model that leverages the robot's current pose and surrounding terrain data as the state space (see Section 3.2) and the front and rear flipper angles as the action space (see Section 3.3). A reward function is established to meet the task's particular requirements (see Section 3.4), and subsequently, the ICM algorithm (see Section 3.5) and a novel DRL network are introduced into the MDP model (see Section 3.6).

### 3.2. State Space

**Local Terrain Information** $H$**:** The reference coordinate system for local terrain information $H$ is denoted as $[\mathcal{L}]$. In this coordinate, the center of the robot chassis serves as the origin, with the X-axis representing the robot's forward direction and the Z-axis indicating the opposite direction of gravity. To effectively express the Robot Terrain Interaction (RTI), we divide the point set $\mathcal{T}$ consisting of terrain point clouds $l(x_T, z_T)$ in front of, behind, and below the robot into N equally spaced subpoint sets $\mathcal{T}_i$, and obtain N average heights $\overline{h_i}$ as local terrain information representation by downsampling (as shown by the red dots in the Figure 3):

$$H = \{\overline{h_i}\} = \{\underset{l(x_T, z_T) \in \mathcal{T}_i}{\text{mean}}(z_T)\},$$

$$\mathcal{T} = \{\mathcal{T}_i\}, \ i = 1 \cdots N, \tag{1}$$

$$x_T \in [x_{T_r}, x_{T_f}] = [-\frac{N}{2}d, \frac{N}{2}d],$$

where $x_T$ and $z_T$ represent the horizontal and vertical coordinates of terrain points in the coordinate system $[\mathcal{L}]$. $x_{T_r}$ and $x_{T_f}$ represent the boundaries of the perception domain along the X-axis and cover a range of $[-\frac{N}{2}d, \frac{N}{2}d]$. Figure 3 depicts the average height $\overline{h}_2$ within subpoint set $\mathcal{T}_2$.

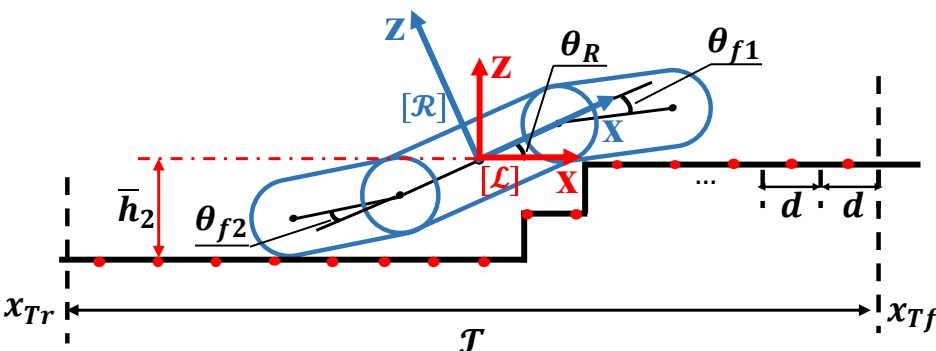

**Figure 3.** State Space: Includes dynamic interactive information between local terrain and robot. The red dots represent the terrain points after downsampling in the local perception domain $\mathcal{T}$.

**Robot State** $E$**:** The coordinate system for the rescue robot is defined as an $[\mathcal{R}]$ coordinate system with the center of the robot chassis as the origin, the X-axis facing the chassis, and the Z-axis perpendicular to the chassis facing upward, as shown in Figure 3 denoted as blue. The robot state $E$ consists of the angle of the robot's front flipper $\theta_{f1}$, the angle of the robot's rear flipper $\theta_{f2}$, and the chassis pitch angle $\theta_R$. The angle of the flipper is the X-axis angle between the flipper and the robot $[\mathcal{R}]$ coordinate system, which is positive if the flipper is above the chassis. The elevation angle of the chassis is the angle between the $[\mathcal{R}]$ coordinate system and the X-axis of the $[\mathcal{L}]$ coordinate system, which is positive if the chassis is above the X-axis of the $[\mathcal{L}]$ coordinate system.

$$E = \{\theta_{f1}, \theta_{f2}, \theta_R\},$$
$$\theta_{f1}, \theta_{f2}, \theta_R \in [-\frac{\pi}{3}, \frac{\pi}{3}]. \tag{2}$$

### 3.3. Action Space

The robot's action, which governs its movement, is generated as the output of our RL network. The tracked robot can adjust its posture by rotating its flipper, facilitating efficient traversal over obstacles. The action space of the MDP model is designed in the form of discrete angular increments of $\Delta\theta_f$ when the flipper rotates, where $\Delta\theta_f = \frac{\pi}{12}$. The front and rear flippers have three motion elements: clockwise rotation of $\Delta\theta_f$, counterclockwise rotation of $\Delta\theta_f$, and nonrotation. Therefore, the motion space is expressed as the combination of nine motions $a$ of the front and rear flipper:

$$A = \{a_{ij}\} = \{i\Delta\theta_f, j\Delta\theta_f\}, \tag{3}$$

$$i = \begin{cases} -1 & \text{front flippers rotate clockwise} \\ 0 & \text{front flippers hold on} \\ 1 & \text{front flippers rotate counterclockwise} \end{cases}$$

$$j = \begin{cases} -1 & \text{rear flippers rotate clockwise} \\ 0 & \text{rear flippers hold on} \\ 1 & \text{rear flippers rotate counterclockwise} \end{cases}.$$

### 3.4. Reward Function

A well-designed reward function is crucial in accomplishing specific tasks for robots [28], as it encourages learning efficient control strategies for the flippers. We merge prior knowledge from human operational experts with quantitative metrics to design reward functions for RTI that satisfy the requirements for smooth and safe obstacle crossing. Specifically, a

front flipper motion reward $R_{flipper}$, a smoothness reward $R_{pitch}$, and a contact stability reward $R_{contact}$ are included.

**Reward of flipper $R_{flipper}$:** The front flipper of the robot plays a crucial role in adjusting its posture while enabling the main track to conform to the terrain as much as possible, and it needs to anticipate upcoming obstacles. We refined the practical flipper strategy based on the manual teleoperation experience introduced by Okada et al. [8] and designed a motion-based reward function specifically for the front flipper. We denote the reward of front flippers as $R_{flipper}$.

Figure 4a,b are schematic diagrams of the reward design of the front flipper when the robot goes up and down obstacles, respectively. In this section, the hinge point $p_{bf}$ between the front flipper and the chassis is selected as the reference point, which is connected with the expanded terrain point (obtained by the original terrain point expanding the half of thickness $B$ of the robot, shown as the red triangle) in the point set $\mathcal{T}_f$ as the vector $\overrightarrow{p_{bf}p_i}$ (shown as the green vector), and the angle between $\overrightarrow{p_{bf}p_i}$ and the X-axis of robot coordinate system are calculated. The one with the largest angle value is selected as the candidate angle of the front flipper (shown by the yellow vector). The $R_{flipper}$ is mainly responsible for guiding the robot to change its posture to adapt to the terrain actively. A proper reward value is conducive to reducing the little and meaningless action exploration of the robot during training and guiding the robot to explore the reasonable front flipper action more efficiently. The absolute difference $\Delta\theta_{f1}$ between the robot's front flipper angle $\theta_{f1}$ and the candidate angle $\theta_{f1}^*$ is taken as the reward index, and $R_{flipper}$ is defined as

$$R_{flipper} = \begin{cases} -1, & \text{if } \Delta\theta_{f1} > \frac{1}{\lambda_1} \\ -\lambda_1\Delta\theta_{f1}, & \text{otherwise} \end{cases},$$

$$\Delta\theta_{f1} = |\theta_{f1} - (\theta_{f1}^* \pm \frac{\pi}{36})|,$$

(4)

where $\lambda_1$ is the threshold coefficient of $\Delta\theta_{f1}$.

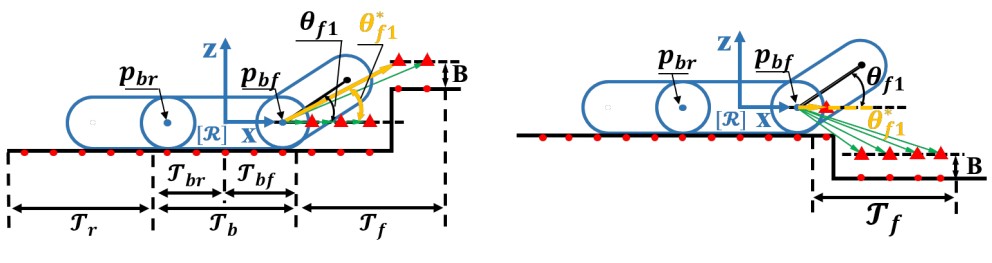

(**a**) Ascending an obstacle.　　　　　　　　　(**b**) Descending an obstacle.

**Figure 4.** Schematic diagram of reward $R_{flipper}$ calculation.

**Reward of Smoothness $R_{pitch}$:** Terrain traversal smoothness is an important evaluation standard, and the pitch angle of the robot chassis changes as gently as possible through the cooperation of the rear and front flippers. We propose to use the relevant indicators of the robot chassis pitch angle as a reward to optimize the robot's terrain traversal stability, denoted as $R_{pitch}$.

The absolute change in pitch angle is $\Delta|\theta_R(t)|$, and the average change in pitch angle in $k$ time steps is defined as $\Delta\theta_r^k(t)$:

$$\Delta|\theta_R(t)| = |\theta_R(t+1)| - |\theta_R(t)|$$

$$\Delta\theta_R^k(t) = \frac{1}{k-1}\sum_{i=t}^{t+k-1}|\theta_R(i+1) - \theta_R(i)|,$$

(5)

where $t$ represents the number of steps the robot performs in a single terrain traversal episode, as shown in Figure 5; the absolute change in pitch angle $\Delta|\theta_R(t)|$ reflects that the pitch change trend of the robot chassis is rising ($\Delta|\theta_R(t)| > 0$), and we limit the situation that the robot is near the overturning boundary, hoping that the pitching trend of the robot

will not rise further when it is in high overturning risk. The average change in pitch angle within $k$ step $\Delta\theta_R^k(t)$ reflects the stability of the robot's terrain traversal. According to the two related indexes of pitch angle mentioned above, the reward of terrain traversal smoothness is designed as $R_{pitch}$, which is defined as

$$R_{pitch} = \begin{cases} -1, & \text{if } (|\theta_R| > \frac{\pi}{4} \text{ and } \Delta|\theta_R(t)| > 0) \\ -1, & \text{if} \Delta\theta_R^k(t) > \frac{1}{\lambda_2} \\ -\lambda_2\Delta\theta_R^k(t), & \text{otherwise} \end{cases}, \quad (6)$$

where $\lambda_2$ is the threshold coefficient of $\Delta\theta_R^k(t)$.

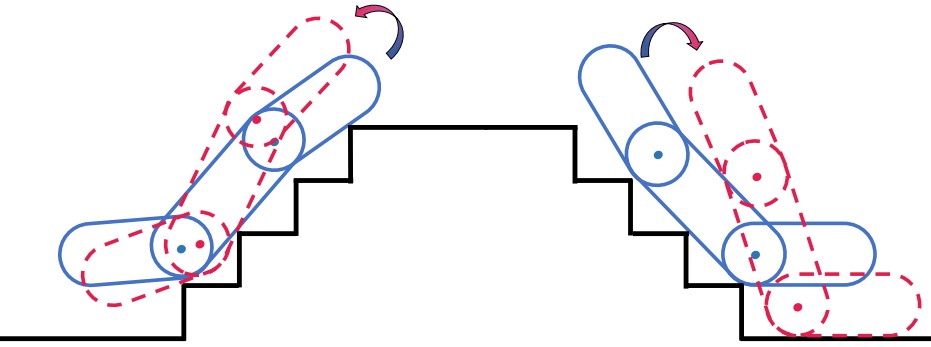

**Figure 5.** Avoidance of pitching dangerous posture. Blue indicates the robot's posture at the current moment, which is at a dangerous edge; Red indicates the possible dangerous trend of the robot, which needs to be avoided.

**Reward of Contact $R_{contact}$:** Under the guidance of $R_{flipper}$ and $R_{pitch}$ rewards, the robot can learn smooth obstacle-crossing maneuvers, but the robot's interaction with the terrain still suffers from some implausible morphology, such as the front and rear flippers supporting the chassis off the ground. This morphology meets the smoothness constraints, but the driving force could be utilized more efficiently, and the torque applied to the flippers needs to be bigger. The contact between the tracks and the terrain is the medium of the robot's driving force. The robot's chassis and flippers are equipped with main and subtracks, respectively, and the size of their contact area with the terrain determines the robot's driving ability, especially the main tracks. Therefore, the $R_{contact}$ reward needs to guide the robot to contact the terrain as much as possible with the chassis tracks to provide sufficient driving force, and there must be contact points at the front and back of the robot's center of mass at the same time.

$$case = \begin{cases} case1, & \text{if } cp_1 \in \mathcal{T}_{br} \text{ and } cp_2 \in \mathcal{T}_{bf} \\ case2, & \text{if } cp_1 \in \mathcal{T}_{br} \text{ and } cp_2 \in \mathcal{T}_f \\ case3, & \text{if } cp_1 \in \mathcal{T}_r \text{ and } cp_2 \in \mathcal{T}_{bf} \\ case4, & \text{otherwise} \end{cases}, \quad (7)$$

$$R_{contact} = \begin{cases} 0, & \text{if } case \in (case1, case2, case3) \\ -1, & case4 \end{cases}. \quad (8)$$

among them, $cp_1$ and $cp_2$ are the farthest contact points between the robot and the terrain; the partitions of $\mathcal{T}_f, \mathcal{T}_{bf}, \mathcal{T}_{br}, \mathcal{T}_r$ are shown in Figure 4a.

**Reward of Terminate $R_{end}$:** In the training process of the RL algorithm, the process from the starting point until the robot meets the end condition is called a terrain traversal episode, and a settlement reward will be given at the end of each episode. When the robot reaches the finish line smoothly, and the chassis is close to the ground, it is regarded as a successful obstacle crossing in this episode, and a big positive reward is obtained. It is necessary to design negative rewards according to the task scene's specific situation to

restrain the robot's dangerous or inappropriate behavior. Thus, when the robot meets the following conditions, it obtains a larger settlement reward and ends the current episode:

$$R_{end} = \begin{cases} +R, & \text{reached} \\ -R, & |\theta_R| \geq \frac{\pi}{3} \\ -R, & t \geq t_{max} \\ -R, & \text{got stuck} \end{cases}. \tag{9}$$

among them, $t_{max}$ represents the maximum number of steps the robot performs in a single terrain traversal episode, and $R$ represents the value of the settlement reward.

In summary, the sum of rewards $R_t^e$ earned by the robot at each obstacle-crossing time $t$ is expressed as Equation (10), where $\kappa$ is the weight of each reward term and $\kappa_1, \kappa_2, \kappa_3 \in [0,1]$.

$$R_t^e = R_{end} + \kappa_1 R_{flipper} + \kappa_2 R_{pitch} + \kappa_3 R_{contact} \tag{10}$$

### 3.5. ICM Algorithm

To improve the robot's exploration of obstacle-crossing action and state, ICM is designed to encourage the robot to try new actions and move to a new state so that it is possible to explore higher-reward obstacle-crossing performance. For the single-step trajectory $\{s_t, a_t, s_{t+1}, R_t\}$ of MDP, ICM is used to evaluate the curiosity level of the robot in the current state. The specific structure is shown in Figure 6, including the Encoder model, Forward prediction model, and Inverse prediction model.

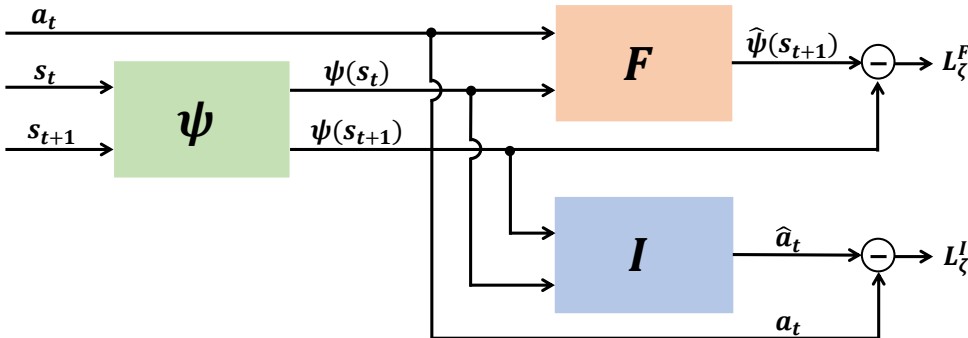

**Figure 6.** ICM Algorithm: The single-step trajectory of MDP serves as inputs, while the $\psi$ and $F$ models compute the curiosity value $L_\zeta^F$ for the present state-action and the $I$ models output loss $L_\zeta^I$.

**Encoder model $\psi$:** Responsible for coding the original state space into a feature space with stronger representation ability. As the key feature state of the obstacle-crossing problem, this feature space should fully contain all the information robots use for decision making. The state space of the traversal MDP model is composed of robot state and terrain information, which fully represents the process of RTI. Encoder model $\psi$ is used to encode current state $s_t$ and transition state $s_{t+1}$ into feature vectors $\psi(s_t)$ and $\psi(s_{t+1})$.

**Forward prediction model $F$:** The $F$ simulates the operation mode of the environmental model and is used to estimate and predict the transfer characteristic state of the robot after acting. Its inputs are robot action $a_t$ and characteristic state vector $\psi(s_t)$, and the predicted transferred characteristic state vector $\hat{\psi}(s_{t+1})$ is the output. The difference between the $\hat{\psi}(s_{t+1})$ output by the $F$ and the characteristic state $\psi(s_{t+1})$ encoded by the encoder $\psi$ indicates the curiosity level of the robot. We introduce the curiosity level as the intrinsic reward $R_t^i$ into the reward function of Reinforcement Learning to encourage robots to explore more unknown states. Thus, the loss function $L_\zeta^F$ of the $F$, the intrinsic curiosity reward $R_t^i$, and the reward sum of each obstacle-crossing time step $R_t$ can be defined as

$$L_\zeta^F = \text{MSELoss}(\hat{\psi}(s_{t+1}), \psi(s_{t+1})), \tag{11}$$

$$R_t^i \triangleq \eta L_\zeta^F = \eta \frac{1}{M} ||\hat{\psi}(s_{t+1}) - \psi(s_{t+1})||_2^2, \tag{12}$$

$$R_t = R_t^e + R_t^i, \tag{13}$$

where $\hat{\psi}(s_{t+1}) = f(\psi(s_t), a_t)$, $\eta$ is the scaling factor of intrinsic curiosity reward, and $M$ is the characteristic state vector $\psi(s_{t+1})$.

**Inverse prediction model** *I*: The core of the *I* is to infer what actions the robot has taken to cause this transfer through the current and transfer states. Its inputs are the current feature state $\psi(s_t)$ and the transition feature state $\psi(s_{t+1})$, and the goal is to output the reasoning action $\hat{a}_t$, which is as small as possible from the action $a_t$ executed by the robot. Under the constraint of the backward inference model, the encoder $\psi$ learns how to extract the accurate feature state space when the network is updated, thus predicting the forward inference model more accurately. The cross-entropy loss function is used to describe the loss of the *I*, which can be expressed as

$$L_\zeta^I = \text{CrossEntropy}(a_t, \hat{a}_t) = -\sum_{i=-1}^{1}\sum_{j=-1}^{1} p(a_{t,ij}) \log(p(\hat{a}_{t,ij})) \tag{14}$$

where $\hat{a}_t = I(\psi(s_t), \psi(s_{t+1}))$; $p(a_{t,ij})$ and $p(\hat{a}_{t,ij})$ represent the probability, receptively, that the front flipper samples the *i* action and the rear flipper samples the *j* action in the vector $a_t$ and $\hat{a}_t$.

At the beginning of the algorithm, the feature extraction ability of the encoder and the prediction ability of the *F* are weak; so, the curiosity value will be large, which drives the robot to explore more. With the progress of obstacle-crossing training, the state of the robot is being explored more and more, and the role of curiosity mechanisms in encouraging exploration will gradually decrease, making the robot make more use of learned action strategies. To sum up, the overall loss function of the ICM algorithm is expressed as follows:

$$L_\zeta = \beta_F L_\zeta^F + \beta_I L_\zeta^I, \tag{15}$$

where $\beta_F$ and $\beta_I$ are the coefficients of forward prediction model loss and backward inference model loss, and $\beta_F, \beta_I \in [0,1]$ and $\zeta$ are the network parameters of ICM.

*3.6. Algorithm and Network*

We show our DRL network architecture in Figure 7, which consists of three main blocks. The block ① highlighted in green is the RTI feature extraction module, which is responsible for generating the interaction features during the terrain traversal process. Terrain data $H$ (green vector) and robot information $E$ (blue vector) are fed into the network to produce the interaction feature vector $S_t^{'}$ via the front-end feature extraction module. Due to the relatively small size of the estimated motion vectors, robot state, and terrain information, there is no requirement for designing complex networks similar to those used for images or 3D point clouds [29]. The network mainly consists of fully connected layers (yellow) and LeakyReLu nonlinear activation layers (orange). Multilayer Perceptron (MLP) is employed to extract features from terrain information $H^{'}$, incorporating them with robot state information $E$ to create a new interactive feature vector $S_t^{'}$ through single-layer MLP.

The block ② highlighted in orange is the D3QN module, which combines the advantages of Double DQN [30] and Dueling DQN [31]. D3QN uses the advantage function $A(S_t^{'}, a_t)$ to evaluate the relative value of each action in the current state to help the robot make more informed decisions. The $A(S_t^{'}, a_t)$ is calculated by subtracting the state value function $V(S_t^{'})$ from the action value function $Q(S_t^{'}, a_t)$ (the output of the Q network). Specifically, the $V(S_t^{'})$ is used to estimate the expected return in a given state, which is to evaluate the value of the $S_t^{'}$. The $Q(S_t^{'}, a_t)$ is used to estimate the expected return of taking a specific action in a given state, which is to evaluate the value of action $a_t$. By calculating the $A(S_t^{'}, a_t)$, we can obtain the potential of each action relative to the average level in the

current state, and the robot can choose the action with the greatest advantage to execute, thus improving the accuracy and effect of decision making.

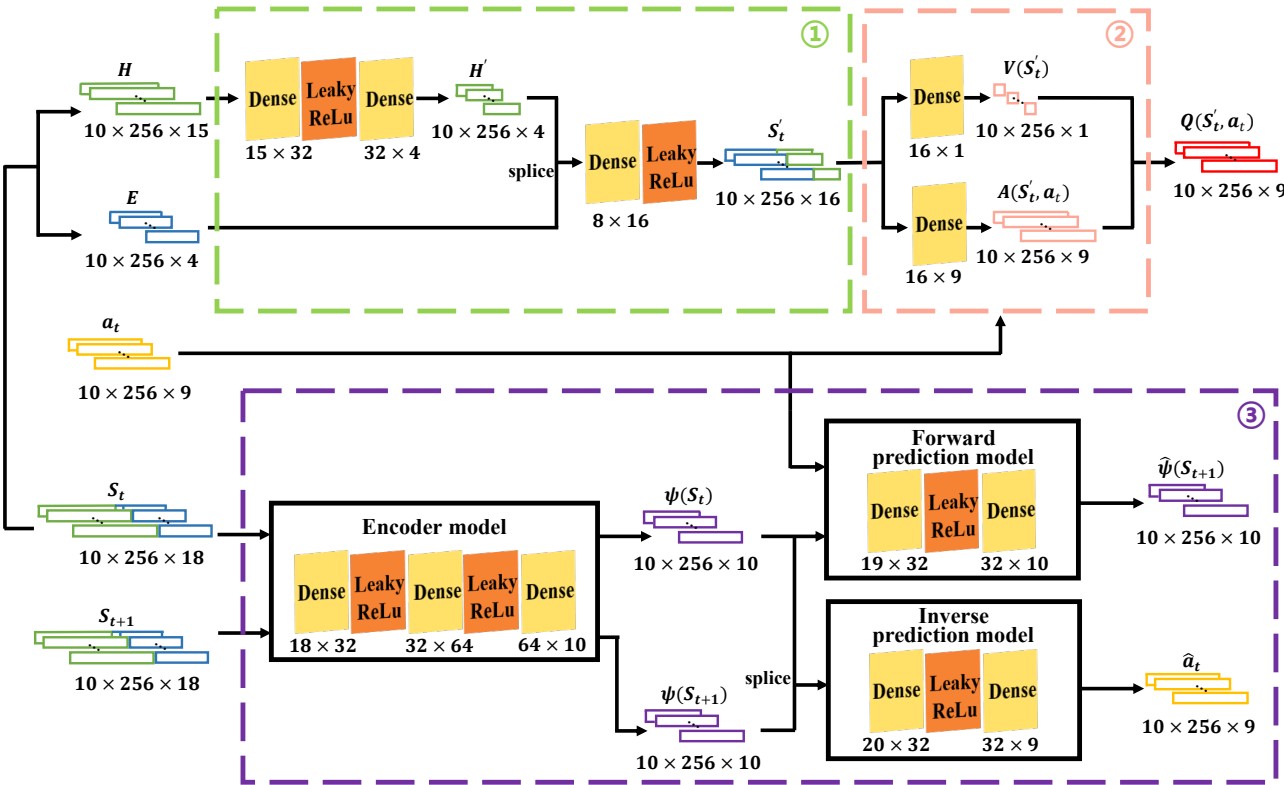

**Figure 7.** Network structure of our ICM-D3QN flipper autonomous control algorithm: ① is a feature extraction and fusion module for extracting and integrating robot and environment state features, ③ is a module for enhancing the efficiency of flipper action exploration, and ② is a deep Q-Learning control generation module.

The block ③ highlighted in purple is the ICM network, which is responsible for outputting the intrinsic reward. The network structure of its Encoder model $\psi$, Forward prediction model $F$, and Inverse prediction model $I$ consists of 3, 2, and 2 MLP layers, respectively.

## 4. Experimental Results

This section mainly demonstrates the training procedure of the proposed ICM-D3QN autonomous flipper control algorithm in the Pymunk simulation and the experimental results in the Gazebo simulation and real environment. Section 4.1 describes the basic settings of the experiments, which specifically include the NuBot-Rescue robot platform and the settings of the training environment and the physical environment. Section 4.2 demonstrates the ablation experiments of the algorithmic network. Section 4.3 demonstrates the algorithm's traversal performance in various simulated and real-world scenarios, including a qualitative analysis of the traversal process, comparing it with the baseline method, and an experiment on urban ruins.

### 4.1. Experiment Setups

#### 4.1.1. Robot Platform

NuBot-Resuce has two main tracks and four independent flippers, presenting a centrosymmetric structure. In addition, its upper level is equipped with a LiDAR sensor (RS-Bpearl is manufactured by RoboSense in Shenzhen, China.), an inertial measurement unit (IMU) sensor (MTi-300 is manufactured by Xsens Technologies in the Netherlands.),

and two cameras for online local mapping, pose estimation, and acquisition of front-to-back images of the robot.

4.1.2. Training Settings

Pymunk (pymunk.org (accessed on 17 September 2023.)) is a simulator based on the Chipmunk physics engine implemented in the Python environment, allowing easy and scalable simulation of physics effects such as rigid bodies, collision detection, and joint constraints. We establish a simulation training environment to expedite the training procedure, leveraging the Pymunk physical simulation engine. The following are the settings of the training environment:

- **Robot Model**. Based on the Section 3.1 assumptions, the articulated tracked robot is modeled as a three-part model based on the symmetric structure, including the rear flippers, the chassis, and the front flippers. Each part comprises a rectangle and two circles, constrained by articulation and rotation relationships.
- **Terrain Model**. To improve the algorithm's ability to generalize, we employ the concept of DR. By creating a range of terrain scenarios with randomized sizes and noise, we aim to enrich the sample experience during training. More specifically, we establish a set of typical urban rescue scenes that include Step, Staircase, and Ramp, as shown as Table 1. We then characterize each scene type using parametric methods. This approach facilitates the organized management of various terrain scenes and enables the generation of terrain shapes with random parameters. The scene is represented using a parameterized format:

$$T = \text{Terrain}(M, X, Z). \tag{16}$$

The interpretation of terrain scene parameters can be observed in Figure 8. In this representation, the step scene consists of a random arrangement of individual step units with varying sizes. The staircase scene consists of a set of step units of the same size, orderly arranged. Similarly, the ramp scene is comprised of ramp units of different sizes. The parameter $M$ signifies the number of units within the scene, while $X$ and $Z$ represent the length height of each unit, respectively. Considering the robot's maximum ability to overcome inclined obstacles at around $45°$, there exists the following constraint for $X_{unit}$ and $Z_{unit}$ in the staircase and ramp terrain:

$$X_{unit} \in [X_{min}, X_{max}], \ Z_{unit} \in [X_{unit}, Z_{max}]. \tag{17}$$

During the algorithm training process, the values of the robot's flipper angle, the parameter values of the terrain shape, and the noise of sensing are randomly generated, which ensures diversity in the robot's state and terrain information, which is beneficial for robustness and adaptability. For each episode of obstacle-crossing training, the start and end points are set to be 1 m away from the start and end of the terrain. The robot is trained through the above three types of scenes, and the parameters used in the training are shown in the Table 2. Finally, the algorithm training was deployed on a desktop computer with a 16-core Intel Core i7-11700F 2.50 GHz CPU, GeForce RTX3060 Ti GPU, and 64 GB RAM.

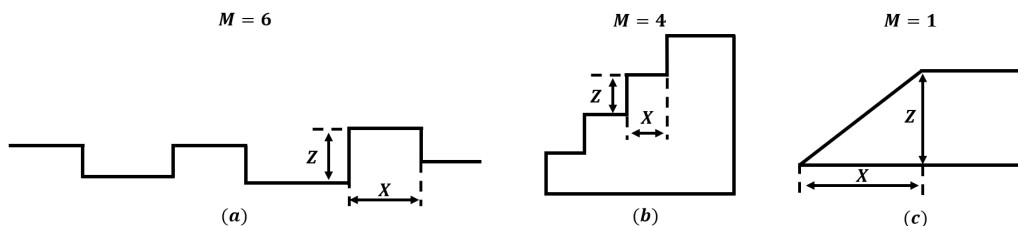

**Figure 8.** The interpretation of terrain scene parameters. (**a**–**c**) separately show a Step composed of 6-step units with different sizes, a Staircase composed of 4-step units with the same size, and a Ramp composed of a ramp unit.

**Table 1.** Parameters of evaluating terrain scenes in training and testing.

| | Training Scene | | | Testing Scene | | | |
|---|---|---|---|---|---|---|---|
| **Scene** | $M$ | $X$/m | $Z$/m | **Scene** | $M$ | $X$/m | $Z$/m |
| Step | $[1,7]$ | $[0.1,1]$ | $[-0.4,0.4]$ | 33.7°-stair | 5 | 0.3 | 0.2 |
| Staircase | $[1,7]$ | $[0.0,0.4]$ | $[0.05,0.25]$ | 45°-stair | 5 | 0.2 | 0.2 |
| Ramp | $[1,3]$ | $[0.6,1.2]$ | $[0.1,1.2]$ | Continuous-ramp | 10 | 0.6 | 0.2 |
| | | | | Urban-ruins | - | 7.5 | 0.7 |

**Table 2.** Parameters of traversal training.

| Parameters | Learning Rate | Replay Buffer Size | Batch Size | $\lambda_1$ | $\lambda_2$ | $\kappa_1, \kappa_2, \kappa_3$ | $\beta_F$ | $\beta_I$ |
|---|---|---|---|---|---|---|---|---|
| Value | $5 \times 10^{-4}$ | $8 \times 10^6$ | 256 | 0.1 | 0.33 | 0.005 | 0.8 | 0.2 |

4.1.3. Testing Settings

In the testing session, the feasibility and generalization ability of the algorithms is evaluated through experiments conducted in Gazebo and the real world.

Gazebo (gazebosim.org (accessed on 17 September 2023)) provides realistic physics-based simulations that allow algorithms to be evaluated in various scenarios and terrains, which helps reduce the risks and costs associated with real-world experimentation. Our robot simulation model references the Contact Surface Motion (CSM) model of Pecka et al. [32]. The simulation parameters of the LiDAR and IMU are consistent with the NuBot-Rescue.

Autonomous navigation and obstacle-crossing experiments occur in unfamiliar surroundings, and a comprehensive comprehension of the environment and the availability of precise maps serve as the bedrock for successful autonomous navigation [33,34]. To ensure the accuracy of the terrain and the robot's pose information, we employ a terrain point cloud map created using the dependable ALOAM map-building algorithm [35,36]. This map, gradually constructed as the robot progresses, gives us 15 downsampled and filtered terrain points and real-time robot states as inputs. The robot maintains a constant speed of 0.15 m/s (the maximum speed can reach 0.3 m/s) without adjusting its forward direction. The algorithm controls the entire flipper with a rotation speed of 25°/s, and it performs five repetitions of the experiment in each experimental scene.

The terrain tested in the experiment includes the 33.7°-stair, 45°-stair, Continuous-ramp, and Urban-ruins, as shown in Figure 9 and Table 1. Specifically, the staircase is the standard terrain for RoboCup RRL rules (rrl.robocup.org (accessed on 17 September 2023)), the Continuous-ramp tests the algorithm's performance in situations with continual pitch bumps, and the Urban-ruins tests the algorithm's ability to adapt to irregular, complex terrain.

Different algorithms govern the robot's traversal of obstacles with varying elapsed durations for obstacle-crossing scenarios of equal length and difficulty. Merely relying on the mean value of the absolute rate of change in the pitch angle, denoted as $|\dot{\theta}_R|$, fails to objectively assess the robot's performance in a single obstacle-crossing task. To address this limitation, we propose the metrics $\hat{\theta}_R$, which combine the task duration and the absolute rate of change in the pitch angle. $\hat{\theta}_R$ represents the integration of the absolute rate of change in the robot's pitch angle over the mission time $t_{cost}$. Meanwhile, $\hat{Z}_R$ denotes the average absolute values of the robot's center height velocity.

$$\hat{\theta}_R = \int_0^{t_{cost}} |\dot{\theta}_R| dt = \sum_{t=1}^{T-1} |\theta_R(t+1) - \theta_R(t)|, \tag{18}$$

$$\hat{Z}_R = \frac{\sum |\dot{z}_R|}{T-1}, \tag{19}$$

where $T$ represents the total number of time steps in the single-trip task, and $\hat{\theta}_R$ and $\hat{Z}_R$ signify overall pitch bump and height shock during the over-the-obstacle endeavor, respectively. A smaller value indicates more seamless control, measured in radians and m/s.

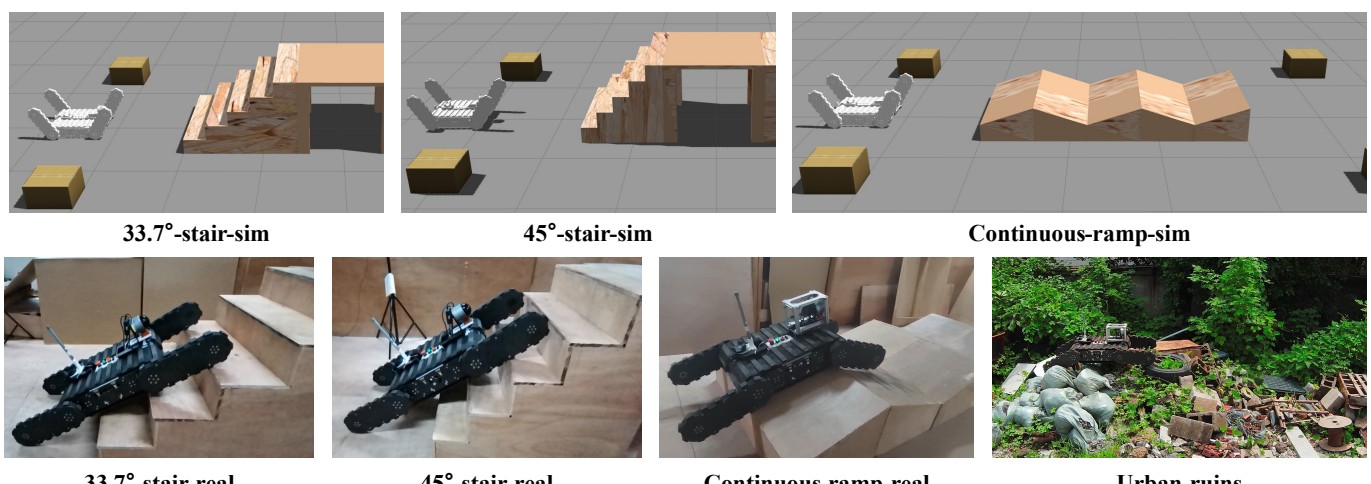

| 33.7°-stair-sim | 45°-stair-sim | Continuous-ramp-sim |

| 33.7°-stair-real | 45°-stair-real | Continuous-ramp-real | Urban-ruins |

**Figure 9.** The testing scenes in simulation and real world, including structured and unstructured and indoor and outdoor situations.

*4.2. Ablation Experiment*

Figure 10 depicts the training progress of the algorithm both before and after incorporating the ICM network structure. The vertical axis represents the average reward per 100 episodes, while the horizontal axis denotes the number of episodes. Upon ICM-D3QN and D3QN initiating training after accumulating $4 \times 10^6$ experiences, ICM-D3QN shows superior convergence performance. This is because DR enhances the generalization ability of the algorithm, but it will lead to higher training costs and difficult convergence. Compared with D3QN, the proposed ICM-D3QN network structure encourages active exploration of unfamiliar states and actions, ultimately leading to stronger convergence ability.

Table 3 illustrates the algorithm's performance in crossing obstacles before and after integrating the ICM network structure under various simulation scenes. To more accurately replicate the potential sensing deviations that may occur in the real environment, a control group called "ICM-D3QN+Noise" was established. Gaussian noise is applied to the sensing data during testing in this group, with $\mu = 0\,\text{cm}$ and $\sigma = 10\,\text{cm}$. The proposed ICM-D3QN flipper control algorithm exhibits smaller $\hat{\theta}_R$ and $\hat{Z}_R$ values than D3QN in the staircase and ramp scenes. It suggests that ICM-D3QN enhances the smoothness of the robot's motion during the obstacle-crossing process. The second column of Table 4 indicates an average enhancement of 13.8% ($\hat{\theta}_R$) and 14.3% ($\hat{Z}_R$) across the three task terrains. Furthermore, the third column of Table 4 demonstrates a slight reduction in smoothness for the ICM-D3QN algorithm in the three task terrains when subjected to higher levels of sensing noise, which is about 6.1% ($\hat{\theta}_R$) and 6.5% ($\hat{Z}_R$). This resilience to terrain-sensing noise highlights the algorithm's robustness.

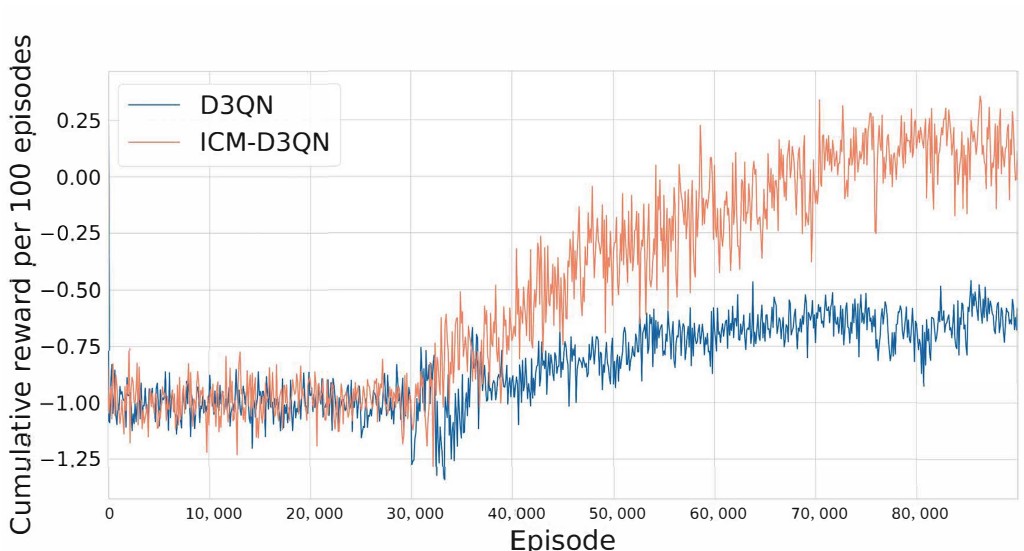

**Figure 10.** Comparison of training curves of D3QN and ICM-D3QN.

To summarize, the ablation experiments confirm the substantial improvements in convergence ability and obstacle-crossing performance achieved by the ICM-D3QN algorithm.

**Table 3.** Ablation study on proposed ICM-D3QN. The bold number indicates the best performance.

| Algorithm | 33.7°-Stair-Sim Ascent and 45°-Stair-Sim Descent | | 45°-Stair-Sim Ascent and 33.7°-Stair-Sim Descent | | Continuous-Ramp-Sim | |
|---|---|---|---|---|---|---|
| | $\hat{\theta}_R$ [rad] | $\hat{Z}_R$ [m/s] | $\hat{\theta}_R$ [rad] | $\hat{Z}_R$ [m/s] | $\hat{\theta}_R$ [rad] | $\hat{Z}_R$ [m/s] |
| D3QN [37] | $5.242 \pm 0.239$ | $0.074 \pm 0.013$ | $5.712 \pm 0.357$ | $0.056 \pm 0.012$ | $4.963 \pm 0.173$ | $0.048 \pm 0.001$ |
| ICM-D3QN + Noise | $4.605 \pm 0.188$ | $0.063 \pm 0.002$ | $5.114 \pm 0.270$ | $0.057 \pm 0.010$ | $4.864 \pm 0.202$ | $\mathbf{0.043 \pm 0.003}$ |
| ICM-D3QN | $\mathbf{4.245 \pm 0.208}$ | $\mathbf{0.057 \pm 0.008}$ | $\mathbf{4.857 \pm 0.179}$ | $\mathbf{0.046 \pm 0.003}$ | $\mathbf{4.603 \pm 0.133}$ | $0.047 \pm 0.002$ |

**Table 4.** The variation range of smoothness metrics of D3QN, ICM-D3QN+Noise compared with ICM-D3QN algorithm in different scenes. $\hat{\theta}_R \uparrow$ and $\hat{Z}_R \uparrow$ represent a worse performance.

| Scenes | D3QN [37] | | ICM-D3QN+Noise | |
|---|---|---|---|---|
| | $\hat{\theta}_R$ [rad] | $\hat{Z}_R$ [m/s] | $\hat{\theta}_R$ [rad] | $\hat{Z}_R$ [m/s] |
| 33.7°-stair-sim ascent and 45°-stair-sim descent | 19.0% | 23.0% | 7.8% | 9.5% |
| 45°-stair-sim ascent and 33.7°-stair-sim descent | 15.0% | 17.9% | 5.0% | 19.3% |
| Continuous-ramp-sim | 7.3% | 2.1% | 5.4% | −9.3% |
| Mean variation range | 13.8%↑ | 14.3%↑ | 6.1%↑ | 6.5%↑ |

### 4.3. Performance Experiment

#### 4.3.1. Qualitative Analysis

This section encompasses experiments conducted on 33.7°-stair, 45°-stair, and Continuous-ramp scenes, aiming to validate the efficacy of transferring the ICM-D3QN from simulation to real-world environments. Figures 11 and 12 show the traversal process in the steep stairs of ICM-D3QN autonomous flipper control algorithm:

- 1∼3: The robot harmonizes its pose with the inclination of ascending stairs through cooperation between the front and rear flippers.
- 4∼6: The robot efficiently ascends the platform, employing active depression of the front and rear flippers to minimize impact and pitch oscillation.
- 7∼9: The algorithm regulates the robot's front flipper to apply significant downward pressure, reducing the descent impact effectively.

- 10~12: The robot adeptly descends the stairs, maintaining stability by using the front flipper for support near ground contact, achieving a smooth arrival at the target.

The traversal process of ICM-D3QN in a Continuous-ramp with symmetrical left and right heights is shown in Figure 13. In the face of the Continuous-ramp with obvious bumps in the pitch angle, the ICM-D3QN algorithm controls the robot's rear flipper to remain flat and controls the front flipper to adapt to the changes in the terrain actively to minimize the degree of the robot's bumps in obstacle crossing, as shown in stages 4~8 of Figure 13.

The aforementioned qualitative analysis evinces the efficacy of our algorithm for seamless migration onto a robotic platform, showcasing smooth and safe obstacle-crossing performance on challenging terrains, such as steep stairs and continuous ramps.

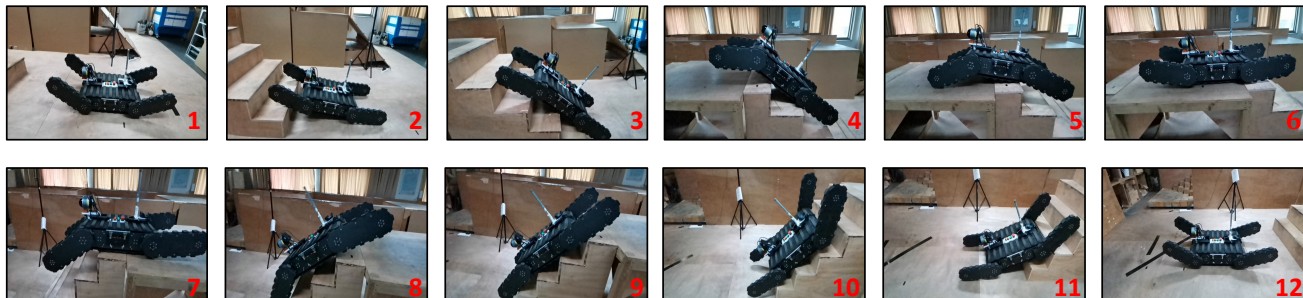

**Figure 11.** Experiment of the ascending 33.7°-stair-real and descending 45°-stair-real.

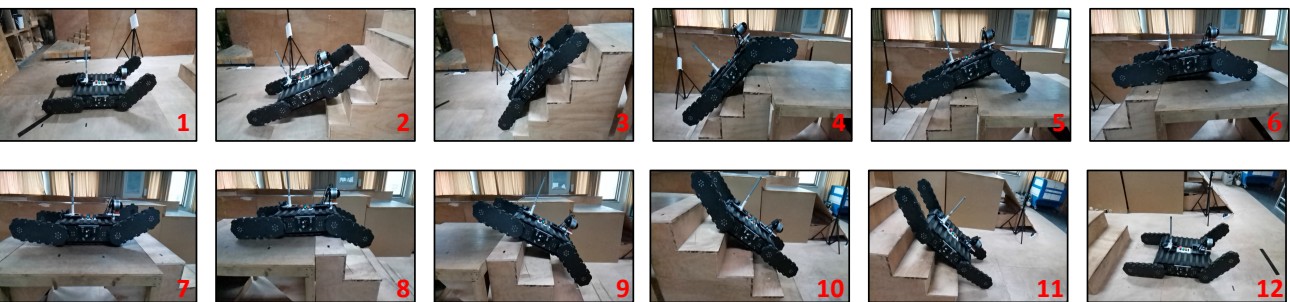

**Figure 12.** Experiment of the ascending 45°-stair-real and descending 33.7°-stair-real.

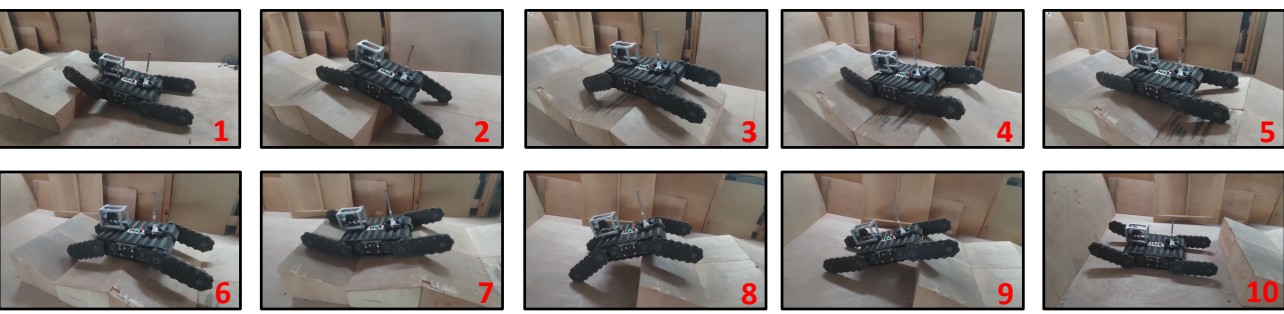

**Figure 13.** The experiment of crossing the continuous ramp-real.

### 4.3.2. Comparison with Baseline Methods

Owing to the profound integration of flipper control methods for articulated tracked robots with the robotic platform, published open-source methodologies for controlling flippers are scarce. A related DRL work uses RGBD images as input and Soft Actor–Critic (SAC) algorithm [38] to train a robot to flip over a staircase scenario in a Gazebo simulation [17]. To ensure comparability, we substitute our robot model for theirs within their open-source Gazebo simulation environment, while the perception method follows the RGBD camera. We then train it using their proposed network architecture and test in 33.7°-stair-sim and 45°-stair-sim scene.

Table 5 demonstrates a comparison of the quantitative metrics' results. Our ICM-D3QN algorithm yields superior outcomes in these metrics, indicating that our approach enables smoother stair ascents and descents by the robot. Moreover, while Mitriakov's method trains separately, with distinct reward functions for ascending and descending stairs; our approach employs a unified set of reward functions to train in more intricate and diverse terrains. As a result, the resulting policy network exhibits enhanced generalization ability and performance.

**Table 5.** Comparison of traversal performance metrics with Mitriakov's method in the simulated stairs. The bold number indicates the best performance.

| Algorithm | 33.7°-Stair-Sim Ascent | | 33.7°-Stair-Sim Descent | | 45°-Stair-Sim Ascent | | 45°-Stair-Sim Descent | |
|---|---|---|---|---|---|---|---|---|
| | $\hat{\theta}_R$ [rad] | $\hat{Z}_R$ [m/s] | $\hat{\theta}_R$ [rad] | $\hat{Z}_R$ [m/s] | $\hat{\theta}_R$ [rad] | $\hat{Z}_R$ [m/s] | $\hat{\theta}_R$ [rad] | $\hat{Z}_R$ [m/s] |
| Mitriakov's | $4.534 \pm 0.248$ | $0.127 \pm 0.009$ | $3.173 \pm 0.180$ | $0.072 \pm 0.003$ | $4.956 \pm 0.135$ | $0.163 \pm 0.023$ | $2.992 \pm 0.077$ | $0.090 \pm 0.003$ |
| Ours | $\mathbf{2.298 \pm 0.223}$ | $\mathbf{0.057 \pm 0.008}$ | $\mathbf{2.012 \pm 0.143}$ | $\mathbf{0.050 \pm 0.003}$ | $\mathbf{2.856 \pm 0.153}$ | $\mathbf{0.062 \pm 0.006}$ | $\mathbf{1.971 \pm 0.070}$ | $\mathbf{0.054 \pm 0.005}$ |

Moreover, we utilize Chen's algorithm [16] to implement a manual modeling method, which involves iteratively exploring the action-pose space and dynamically planning the most cost-effective sequence of actions. This algorithm is deployed on the NuBot-Rescue robot platform and evaluated in a staircase scene. Table 6 demonstrates a comparative analysis of quantitative metrics in a real staircase environment. For the $\hat{\theta}_R$ metric for descending 33.7° stair and the $t_{cost}$ metric for descending 45° stair, Chen's approach achieves more similar results to ours. The overall results highlight the obstacle-crossing performance of our ICM-D3QN algorithm in the face of steep staircases, with higher overall pitch stability, lower center-of-mass height change rate, and shorter obstacle-crossing elapsed time.

**Table 6.** Comparison of traversal performance metrics with Chen's method in real-world stairs. The bold number indicates the best performance.

| Scene | $\hat{\theta}_R$ [rad] | | $\hat{Z}_R$ [m/s] | | $t_{cost}$ [s] | |
|---|---|---|---|---|---|---|
| | Chen's | Ours | Chen's | Ours | Chen's | Ours |
| 33.7°-stair-real ascent | $3.522 \pm 0.274$ | $\mathbf{2.966 \pm 0.196}$ | $0.053 \pm 0.003$ | $\mathbf{0.051 \pm 0.002}$ | $30.074 \pm 2.327$ | $\mathbf{28.550 \pm 2.051}$ |
| 33.7°-stair-real descent | $\mathbf{2.653 \pm 0.638}$ | $2.680 \pm 0.103$ | $0.065 \pm 0.003$ | $\mathbf{0.052 \pm 0.001}$ | $31.178 \pm 4.056$ | $\mathbf{26.680 \pm 0.467}$ |
| 45°-stair-real ascent | $3.853 \pm 0.281$ | $\mathbf{3.429 \pm 0.258}$ | $0.069 \pm 0.004$ | $\mathbf{0.059 \pm 0.037}$ | $29.754 \pm 4.599$ | $\mathbf{28.220 \pm 1.419}$ |
| 45°-stair-real descent | $3.350 \pm 0.241$ | $\mathbf{3.271 \pm 0.102}$ | $0.120 \pm 0.106$ | $\mathbf{0.055 \pm 0.002}$ | $\mathbf{26.449 \pm 2.302}$ | $27.424 \pm 0.492$ |

### 4.3.3. Real-World Urban Searching and Rescuing Experiment

The experimental site of the urban ruin is situated upon an outdoor lawn, predominantly consisting of sandbags, concrete blocks, and abandoned furniture, spanning approximately 7.5 m. The site encompasses asymmetrical obstacles, while surrounding grass influences perception accuracy, rendering it apt for assessing our algorithms' generalization capacity and obstacle-traversing efficacy in intricate, asymmetrical environments. Throughout the experimental trials, the robot's directional movement is directed by a remote human operator utilizing real-time wireless transmission of video imagery, while the ICM-D3QN algorithm governs the manipulation of the flippers.

Figures 14 and 15 showcase the robot smoothly traversing amidst a backdrop of urban ruins, maintaining a stable posture. When the robot ascends a hill constructed with sandbags and concrete blocks, even though there is a bump in the hill (as shown as Figure 14: 2∼3), the algorithm regulates the rear flippers to exert slight downward pressure, thereby preventing any potential tipping of the chassis. Simultaneously, the front flippers exert downward force to minimize any impact when traversing the bump, enabling the robot to safely and smoothly surmount the hill. Furthermore, even when navigating across a pallet characterized by a hollow surface (unrepresented in the training

data), our algorithm successfully guides the robot to maintain an exemplary traversal form, as exemplified in Figure 14: 7∼12 and Figure 15: 1∼5. Video describing more details about obstacle crossing can be shown in Supplementary Materials.

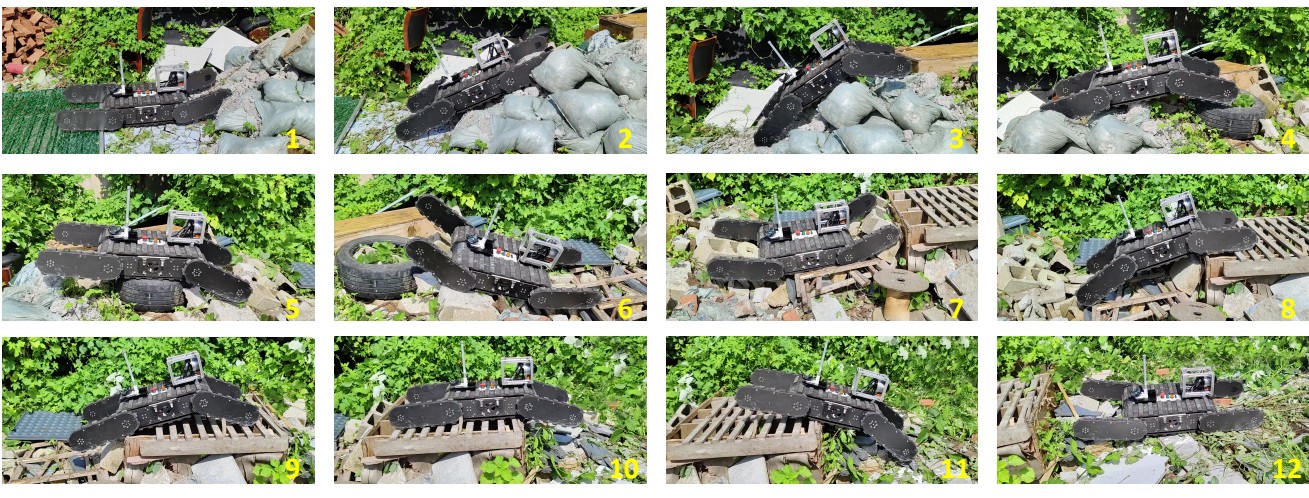

**Figure 14.** Experiment on the urban ruins (forward).

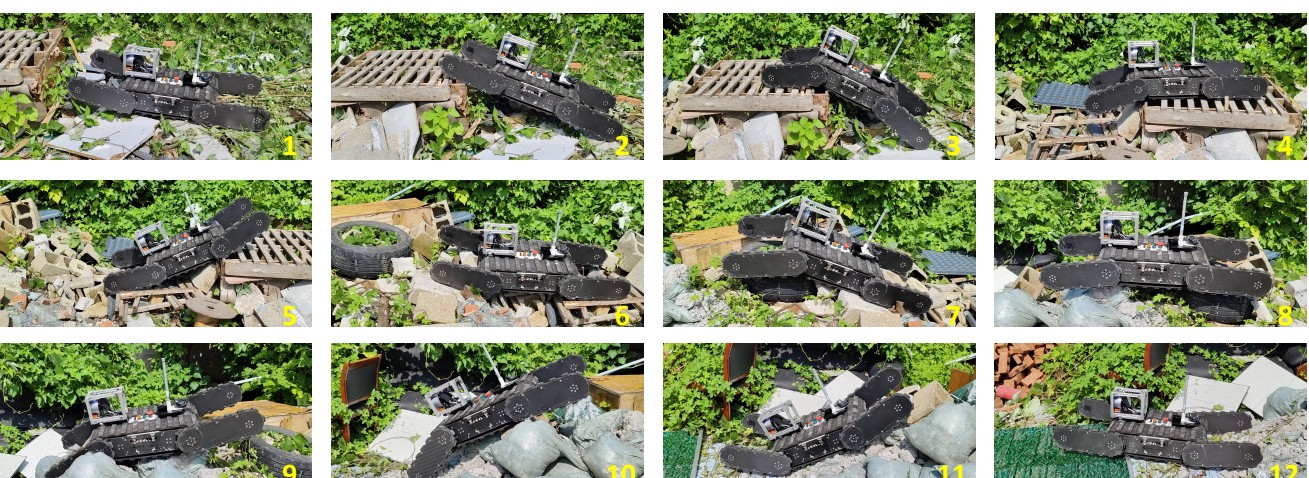

**Figure 15.** Experiment on the urban ruins (backward).

Table 7 presents the mean quantitative data of eight experimental rounds conducted, both in forward and backward directions. In addition to the three metrics listed previously, they record the overall stability of the roll and yaw angles, providing a more comprehensive overview of the algorithm's performance in traversing obstacles within an asymmetrical environment. Backward passes exhibit more pronounced $\hat{\theta}_R$ metrics than forward passes due to the more notable influence of slippery sandbags and moveable concrete blocks on the robot's descent. The experimental results decisively demonstrate our algorithm's resolute generalization ability within asymmetric environments, thereby facilitating the robot's secure, seamless, and expeditious traversal of obstacles encountered in certain scenarios, such as collapsed building sites in urban areas.

**Table 7.** Traversal performance metrics of Urban Searching and Rescuing experiment in the real world.

| Scene | $\hat{\theta}_R$ [rad] | $\hat{\theta}_{roll}$ [rad] | $\hat{\theta}_{yaw}$ [rad] | $\hat{Z}_R$ [m/s] | $t_{cost}$ [s] |
|---|---|---|---|---|---|
| Urban ruins (forward) | $8.011 \pm 0.720$ | $6.997 \pm 0.656$ | $2.619 \pm 0.342$ | $0.059 \pm 0.003$ | $77.562 \pm 4.814$ |
| Urban ruins (backward) | $8.434 \pm 0.377$ | $7.036 \pm 0.713$ | $2.753 \pm 0.277$ | $0.067 \pm 0.007$ | $76.662 \pm 3.097$ |

## 5. Conclusions

This paper introduces a novel flipper control algorithm, ICM-D3QN, based on DRL. It aims to address the challenges faced by tracked robots when crossing obstacles. Our algorithm leverages the state information and local terrain data to learn flipper control actions through an optimized reward function. By incorporating the ICM module and DR technique, we enhance the algorithm's convergence and obstacle-crossing performance, enabling the robot to navigate and overcome obstacles seamlessly. We assess the performance of ICM-D3QN against D3QN in the Gazebo simulation environment. The results indicate that integrating the ICM mechanism improves the algorithm's convergence and obstacle-crossing capabilities. Furthermore, we quantitatively compare ICM-D3QN with Mitrikov's approach and Chen's manual modeling technique in a challenging staircase scenario. The outcome highlights the superiority of our algorithm's learned flipper control strategy across various metrics, including overall pitch angle stability, stability of center-of-mass height change rate, and obstacle-crossing time. Additionally, ICM-D3QN demonstrates certain generalization abilities, maintaining smooth obstacle crossing even in complex and asymmetric urban ruins and resilience in outdoor scenarios with high sensing noise.

Nevertheless, when confronted with environments featuring substantial lateral oscillations, the applicability of the proposed technique may be diminished. In future work, we further deliberate on the intricate challenge of enabling independent flipper locomotion to adequately address the demands of adapting to diverse, rugged terrains.

**Supplementary Materials:** The following supporting information can be downloaded at: https://www.mdpi.com/article/10.3390/rs15184616/s1, Video ICM-D3QN: Deep Reinforcement Learning for Flipper Control of Tracked Robots in Urban Rescuing Environments.

**Author Contributions:** Methodology, H.P.; Software, H.P. and K.H.; Validation, H.P. and B.C.; Formal analysis, H.P. and B.C.; Resources, K.H.; Writing—original draft, H.P. and X.C.; Writing—review & editing, H.P., X.C. and J.R.; Supervision, X.C., J.R. and H.L.; Project administration, H.Z. and H.L. All authors have read and agreed to the published version of the manuscript.

**Funding:** This research was funded by the National Science Foundation of China under Grant U1913202, as well as Major Project of Natural Science Foundation of Hunan Province under Grant 2021JC0004.

**Data Availability Statement:** The data supporting this study's findings are available from the corresponding author or the first author upon reasonable request.

**Conflicts of Interest:** The authors declare no conflict of interest.

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
