# Peer review of "Deep Reinforcement Learning for Flipper Control of Tracked Robots in Urban Rescuing Environments"

_remotesensing, doi:10.3390/rs15184616_

Round 1

Reviewer 1 Report

This paper utilizes DRL to address the autonomous flipper control problem of tracked robots in the urban rescuing environment, which has important application value for post-disaster rescue in real life.. Compared with existing model-based control approaches, their main contributions include a feature extraction module for extracting and integrating robot and environment state features, and a curiosity module for enhancing the efficiency of flipper action exploration.  In addition, a specific reward function is designed, considering both safety and smoothness. Overall, the idea in this paper is interesting. The validation results on the simulator and in the real world look promising. However, the technical rigor should be strengthened, and the writing sometimes has unclear descriptions.

Suggestions to improve the paper:

1. Are there any other solutions and references for solving this kind of task by means of RL?

2. What the red line in Figure 1 (a) represents should be explained in the caption.

3. What is the definition of CrossEntropy( ) in paragraph 1 on page 10?

4. What is the definition of the advantage function A( , ) in paragraph 5 on page 10? 

5. Why are the flippers of the robot symmetrical between left and right during its movement? And what are the advantages and limitations of this? The above questions should be explained in the text.

6. It would be better to reorganize Section 1 and Section 2. Please describe the contribution in a more systematic way.

7. The meaning of the special markings in the illustrations should be explained in the caption. For example, what do the red points mean in Figure 2 and Figure 3?

8. Abbreviated words in the paper should be defined at their first occurrence. For example, the definition of RTI should be given in the third paragraph on page 5.

9. The hyperparameters of the proposed method, such as learning rate, $\beta_F$, $\beta_I$ should be listed.

10. The initial letter of the word corresponding to the acronym USAR should be capitalized.

11. On page 1, there is an incorrectly labeled reference after “rugged terrains”

12. On page 4, there is an incorrectly labeled reference after “set the goal of exploration

Attention should be paid to the standardization of English expression.

Author Response

Response to Reviewer 1 Comments

We would like to thank the editor and all the reviewers for all the comments and suggestions. The feedback has been constructive and allowed us to address shortcomings in the paper effectively. We have addressed the issues raised by the reviewers and incorporated their suggestions in the new version of the paper.  

All the changes are highlighted in the revised version. We provide detailed responses to all the comments raised by reviewer 1 in an item-by-item manner as follows:

Comments 1: Are there any other solutions and references for solving this kind of task by means of RL?

Response 1: Thanks for your advice. We highlight other RL solutions in the related work (lines 86-112). Paolo et al. initially employed a comprehensive end-to-end DRL approach to address the challenge of autonomous flipper control. Works by Mitriakov et al. optimized the mechanical arm's and chassis's overall stability by incorporating it into the reward function. Zimmermann et al. used real terrain traversal data as the expectation, and the feature with the smallest residual in Robot Terrain Interaction (RTI) was extracted by DL. The Q-learning method was then used to learn the strategy of switching among five predefined flippers from the feature.

Comments 2: What the red line in Figure 1 (a) represents should be explained in the caption. 

Response 2: Thanks for your advice. The red line in Figure 1 (a) represents the local enlarged view of the robot crossing obstacles and we highlight its description in the heading of Figure 1.

Comments 3: What is the definition of CrossEntropy( ) in paragraph 1 on page 10?

Response 3: Thanks for your advice. We add the definition and description of CrossEntropy on page 10, lines 302-305.

Comments 4: What is the definition of the advantage function A( , ) in paragraph 5 on page 10?  

Response 4: Thanks for your advice. We add the definition and description of advantage function on page 10, lines 326-336. 

Comments 5: Why are the flippers of the robot symmetrical between left and right during its movement? And what are the advantages and limitations of this? The above questions should be explained in the text. 

Response 5: Thanks for your advice and question. To reduce the sideslip of the robot, the left and right flippers usually work synchronously in practical use. However, this assumption weakens the robot's adaptability to the terrain to a certain extent, for example, in an environment with great differences in left and right heights. Therefore, this work is still carried out in an environment with large vertical fluctuations, and it can also adapt well to some unstructured environments with moderate left and right fluctuations. The work of completely independent movement of flippers will be carried out in the future. We highlight why flippers are synchronized left and right in Section 3.1, lines 164-17 and the advantages and limitations of this assumption in Section 5, lines 544-547.

Comments 6: It would be better to reorganize Section 1 and Section 2. Please describe the contribution in a more systematic way.

Response 6: Thanks for your advice. We have revised and systematically described our contribution and highlight it in Section 1, lines 44-71.

Comments 7: The meaning of the special markings in the illustrations should be explained in the caption. For example, what do the red points mean in Figure 2 and Figure 3? 

Response 7: Thanks for your advice. We highlight the meaning of the red terrain points in the heading of Figure 3 and lines 186-187.

Comments 8: Abbreviated words in the paper should be defined at their first occurrence. For example, the definition of RTI should be given in the third paragraph on page 5.

Response 8: Thanks for your advice. We highlight the abbreviated words (Robot Terrain Interaction , RTI) in line 183.

Comments 9: The hyperparameters of the proposed method, such as learning rate, $\beta_F$, $\beta_I$ should be listed. 

Response 9: Thanks for your advice. We add and highlight the hyperparametric values of algorithm training in page 12, Table 2.

Comments 10: The initial letter of the word corresponding to the acronym USAR should be capitalized. 

Response 10: Thanks for your advice and we have checked and revised the abbreviations of the full text.

Comments 11: On page 1, there is an incorrectly labeled reference after “rugged terrains” 

Response 11: Thanks for your advice and we have checked and revised the references in the full text. We highlight the correction in line 20.

Comments 12: On page 4, there is an incorrectly labeled reference after “set the goal of exploration” 

Response 12: Thanks for your advice and we have checked and revised the references in the full text. We highlight the correction in line 132-133.

Reviewer 2 Report

In line 427-429, “After conducting multiple experiments, we determine that collecting 80,000 episodes of initial experience yielded optimal results.”The description is unclear and accurate, and 80000 cannot be derived from Figure 10. Please provide further description and explanation.

It is recommended to supplement the point cloud data collected by LiDAR when making control decisions in Figures 14 and 15, and provide explanations.

Intrinsic Curiosity Module (ICM) was not explained in the position where the paper first appeared, and no corresponding literature appeared on line 41.

In line 185, no explanation for RTI was provided.It is recommended to supplement the environmental information data images perceived by radar in Figures 14 and 15.

Author Response

Response to Reviewer 2 Comments

We would like to thank the editor and all the reviewers for all the comments and suggestions. The feedback has been constructive and allowed us to address shortcomings in the paper effectively. We have addressed the issues raised by the reviewers and incorporated their suggestions in the new version of the paper.  

All the changes are highlighted in the revised version. We provide detailed responses to all the comments raised by reviewer 2 in an item-by-item manner as follows:

Comments 1: In line 427-429, “After conducting multiple experiments, we determine that collecting 80,000 episodes of initial experience yielded optimal results.”The description is unclear and accurate, and 80000 cannot be derived from Figure 10. Please provide further description and explanation.

Response 1: Thanks for your advice. I'm sorry that this is a wrong description in our writing. We want to express that after many experiments, the value of the hyperparameter, the size of the replay buffer, is confirmed as 8e6, which is not 80,000 episodes as stated in the original text. We have highlighted and listed the value of this parameter in Table 2 of the training hyperparameter. In addition, we revise and highlight the original wrongly described sentence on page 14, lines 430-435.

Comments 2: It is recommended to supplement the point cloud data collected by LiDAR when making control decisions in Figures 14 and 15, and provide explanations.

Response 2: Thank you for your advice. In Figure 1c, we present the global map in the USAR environment. Considering the article's length, we have added additional point cloud images in Figures 14 and 15. In addition, the video we submitted has real-time point cloud images of robots crossing obstacles in the USAR environment. Therefore, in the end, we didn't further present the point cloud in Figures 14 and 15. Thank you again for your advice, and I greatly hope to get your understanding.

Comments 3: Intrinsic Curiosity Module (ICM) was not explained in the position where the paper first appeared, and no corresponding literature appeared on line 41.

Response 3: Thanks for your advice. We add and highlight the corresponding literature citations of ICM.

Comments 4: In line 185, no explanation for RTI was provided.It is recommended to supplement the environmental information data images perceived by radar in Figures 14 and 15.

Response 4: Thanks for your advice. We highlight the abbreviated words (Robot Terrain Interaction , RTI) in line 183.

Reviewer 3 Report

The authors present a deep reinforcement learning approach to autonomous positioning of the flippers of a rescue robot.  They present a novel architecture for the network, which includes an enhancement of a curiosity module to attempt to ensure sufficient coverage of the action space during training.  A simulation is used to learn a mapping between environmental terrain, chassis pose, and desired flipper angles.  The learned policy is then evaluated in a higher-fidelity simulator and several real-world environments.  It is compared against two other flipper policies in order to show improvement.

The paper is generally well-written, with some minor grammar and spelling errors.  (e.g. stucked, 265; inappropriate flippers, 84; our proposal posit 104, poses estimation 349 are a few that jumped out at me).

Pymunk needs to be explained earlier in the paper, as it’s referenced several times before the description.  Some acronyms weren’t defined before first use (e.g. RTI).

The description of the reward functions used are well-motivated and clear, with one exception - how the ‘expanded terrain point’ is chosen isn’t clear in 3.4, under R_flipper.  The text points to the green arrow in Figure 4, but there are two in Figure 4b, and mentions the largest angle, but $$\theta^*_{f1}$$ isn’t the largest angle with respect to the chassis or previous flipper position.

The ablation experiment should be motivated a bit better for those of us not working with DRL networks - it’s necessity and conclusions are not clear (to me).  The performance experiment is clear.  A reference to (18) and (19) in the results section would have saved me some searching for the definitions of the metrics used.

The paper is generally well-written, with some minor grammar and spelling errors.  (e.g. stucked, 265; inappropriate flippers, 84; our proposal posit 104, poses estimation 349 are a few that jumped out at me).

Author Response

Response to Reviewer 3 Comments

We would like to thank the editor and all the reviewers for all the comments and suggestions. The feedback has been constructive and allowed us to address shortcomings in the paper effectively. We have addressed the issues raised by the reviewers and incorporated their suggestions in the new version of the paper.  

All the changes are highlighted in the revised version. We provide detailed responses to all the comments raised by reviewer 3 in an item-by-item manner as follows:

Comments 1: The paper is generally well-written, with some minor grammar and spelling errors.  (e.g. stucked, 265; inappropriate flippers, 84; our proposal posit 104, poses estimation 349 are a few that jumped out at me). Pymunk needs to be explained earlier in the paper, as it’s referenced several times before the description.  Some acronyms weren’t defined before first use (e.g. RTI).

Response 1: Thanks for your advice. We have checked, corrected and highlighted the errors in English expression and abbreviated words (Robot Terrain Interaction , RTI) mentioned above and in the full text. In addition, we adjust the appearance of the word ‘Pymunk’ in the experimental setting to avoid premature explanation.

Comments 2: The description of the reward functions used are well-motivated and clear, with one exception - how the ‘expanded terrain point’ is chosen isn’t clear in 3.4, under R_flipper.  The text points to the green arrow in Figure 4, but there are two in Figure 4b, and mentions the largest angle, but $$\theta^*_{f1}$$ isn’t the largest angle with respect to the chassis or previous flipper position.

Response 2: Thank you for your advice. I'm sorry there is an error in Figure 4(b). The maximum expected angle of flippers $\theta*_{f1}$ defined by us is regardless of the absolute value. We modified and highlighted Figure 4(b) on page 7 and added the expression "extended topographic points" in lines 218-219.

Comments 3: The ablation experiment should be motivated a bit better for those of us not working with DRL networks - it’s necessity and conclusions are not clear (to me).  The performance experiment is clear.  A reference to (18) and (19) in the results section would have saved me some searching for the definitions of the metrics used.

Response 3: Thanks for your advice. Our ablation experiment is mainly to verify the improvement of the original D3QN obstacle-crossing algorithm after adding the ICM module. On the one hand, we verify the improvement of obstacle-crossing performance; on the other hand, we verify the reduction of the sensitivity to perceived noise. Specifically, both D3QN and ICM-D3QN add some perceptual noise interference during training. The ablation experiment proves the algorithm is more robust to noise after adding the ICM module. Table 3 is the numerical comparison of the algorithm's performance in three different terrains before and after improvement, and Table 4 is the intuitive presentation of the improvement range. Regarding the indexing of Formula (18) and Formula (19), I'm sorry I didn't find a suitable way to realize it. I tried \cref{eq:18} to refer to it, but it would make the table too large, so I didn't make any adjustments, and I hope to get your understanding.
